# Species and drug susceptibility profiles of staphylococci isolated from healthy children in Eastern Uganda

David Patrick Kateete[1,2]*, Benon B. Asiimwe[2], Raymond Mayanja[2,3], Christine Florence Najjuka[2], Elizeus Rutebemberwa[4]

**1** Department of Immunology and Molecular Biology, School of Biomedical Sciences, Makerere University College of Health Sciences, Kampala, Uganda, **2** Department of Medical Microbiology, School of Biomedical Sciences, Makerere University College of Health Sciences, Kampala, Uganda, **3** Makerere University Walter Reed Project, Kampala, Uganda, **4** Makerere University School of Public Health, Kampala, Uganda

* dkateete@chs.mak.ac.ug

**Data Availability Statement:** All relevant data are within the manuscript and its Supporting Information files.

## Abstract

Staphylococci are a key component of the human microbiota, and they mainly colonize the skin and anterior nares. However, they can cause infection in hospitalized patients and healthy individuals in the community. Although majority of the *Staphylococcus aureus* strains are coagulase-positive, some do not produce coagulase, and the isolation of coagulase-positive non-*S. aureus* isolates in humans is increasingly being reported. Therefore, sound knowledge of the species and characteristics of staphylococci in a given setting is important, especially isolates from children and immunocompromised individuals. The spectrum of Staphylococcus species colonizing children in Uganda is poorly understood; here, we aimed to determine the species and characteristics of staphylococci isolated from children in Eastern Uganda. Seven hundred and sixty four healthy children less than 5 years residing in Iganga and Mayuge districts in Eastern Uganda were enrolled. A total of 513 staphylococci belonging to 13 species were isolated from 485 children (63.5%, 485/764), with *S. aureus* being the dominant species (37.6%, 193/513) followed by *S. epidermidis* (25.5%, 131/513), *S. haemolyticus* (2.3%, 12/513), *S. hominis* (0.8%, 4/513) and *S. haemolyticus/lugdunensis* (0.58%, 3/513). Twenty four (4.95%, 24/485) children were co-colonized by two or more Staphylococcus species. With the exception of penicillin, antimicrobial resistance (AMR) rates were low; all isolates were susceptible to vancomycin, teicoplanin, linezolid and daptomycin. The prevalence of methicillin resistance was 23.8% (122/513) and it was highest in *S. haemolyticus* (66.7%, 8/12) followed by *S. aureus* (28.5%, 55/193) and *S. epidermidis* (23.7%, 31/131). The prevalence of multidrug resistance was 20.3% (104/513), and 59% (72/122) of methicillin resistant staphylococci were multidrug resistant. Four methicillin susceptible *S. aureus* isolates and a methicillin resistant *S. scuiri* isolate were mupirocin resistant (high-level). The most frequent AMR genes were *mecA*, *vanA*, *ant(4')-Ia*, and *aac(6')-Ie- aph(2'')-Ia*, pointing to presence of AMR drivers in the community.

**Funding:** The authors received no specific funding for this work.

**Competing interests:** The authors have declared that no competing interests exist.

## Introduction

Staphylococci are a key component of the human microbiota, and around 47 species have been identified [1, 2]. They mainly colonize the skin, anterior nares, the nasopharynx, and gut [2–4]. Staphylococci are generally nonpathogenic however, when there is breach of the body barriers (e.g. damage to the skin and mucous membranes) or when the immune system is weakened, they become opportunistic pathogens and cause a variety of mild to severe community- and hospital-acquired infections [2–4].

One of the most useful characteristics used to differentiate staphylococci is their ability to produce coagulase. In this regard, staphylococcal isolates are broadly classified into two groups–the coagulase-positive staphylococci (CoPS) comprising of mainly *Staphylococcus aureus*, which is more pathogenic, and the coagulase-negative staphylococci (CoNS) that are less pathogenic but include majority of the species [2]. While majority of *S. aureus* strains are overwhelmingly coagulase-positive, some are atypical in that they do not produce coagulase [5–11]. Among the CoNS, the medically relevant species include *S. epidermidis*, *S. haemolyticus*, *S. hominis* and *S. saprophyticus* [1, 2]. Interestingly, one species, *Staphylococcus schleiferi*, includes both CoNS and CoPS subspecies i.e. *S. schleiferi* subsp. *schleiferi* and *S. schleiferi* subsp. *coagulans*, respectively [2]. Furthermore, there are five additional CoPS species i.e. *S. delphini*, *S. hyicus*, *S. intermedius*, *S. intrae* and *S. pseudintermedius*, which are typically animal-associated [2] but they have also been isolated from human samples. Although these non-*S. aureus* CoPS species are characteristically animal-associated [2], they are likely to be prevalent in human specimens in the developing countries [5] where rural, agrarian populations predominate.

Staphylococcal colonization is a known risk factor for staphylococcal disease [12]. Globally, there is an increase in prevalence of nosocomial infections caused by CoNS and atypical *S. aureus*, and this is attributed to the increase in use of procedure-related devices in human medicine [2]. CoNS and atypical *S. aureus* infections are difficult to treat as they are more likely to be methicillin-resistant and multidrug-resistant; moreover, CoNS species like *S. lugdenensis* and *S. saprophyticus* are associated with severe infections i.e. infective endocarditis and urinary tract infects, respectively [2]. It is therefore important to unambiguously identify staphylococci to species-level, particularly isolates from vulnerable groups such as children and immunocompromised individuals. Additionally, an understanding of the dynamics of staphylococcal colonization in a given setting, especially methicillin resistant CoNS (MRCoNS), is critical as they are considered to be the reservoir of antibiotic resistance genes for the more pathogenic species (*S. aureus*) [3, 13]. Several studies have reported transfer of the staphylococcal cassette chromosome *mec* (SCC*mec*), a transposon that bears the molecular determinant of methicillin resistance (*mecA* gene), between *S. epidermidis* and *S. aureus* [14]. While the role of CoNS as reservoirs of the SCC*mec* element is well-established, the spectrum of species potentially involved in SCC*mec* exchange is not well documented, particularly in the developing countries. Here we aimed to determine the species and characteristics of staphylococci isolated from children in Eastern Uganda. As virulence and antimicrobial resistance (AMR) tend to coexist in staphylococci [15, 16], we also determined the frequency of genes associated with these phenotypes among community-associated staphylococci in Eastern Uganda.

## Materials and methods

### Ethics statement

This study was approved by the Institutional Review Boards of the Schools of Medicine, Public Health, and Biomedical Sciences at Makerere University College of Health Sciences (REF

2011–183; SBS194), and by the Uganda National Council for Science and Technology (Ref HS 1080) [17–20]. Written informed consent was obtained from the parents/guardians of the children, and the consent procedure included consent for storage of samples for further studies. Since the study was conducted on stored isolates, the Research Ethics Committees waived the need for re-consenting of the participants.

## Study setting, samples and isolates

This cross-sectional study was nested in previous studies/projects that investigated the pneumococcal nasopharyngeal colonization and integrated community case management of malaria and pneumonia in children less than 5 years of age at the Iganga/Mayuge Health and Demographic Surveillance Site (IMHDSS) in rural Eastern Uganda [20, 21]. The study population was described before [20–22]; briefly, 764 healthy children less 5 years at the IMHDSS were enrolled and sampled. Following consent, a study nurse collected two nasopharyngeal samples (swabs) from each child. Swabs were transported in small batches to the Clinical Microbiology laboratory in Kampala for culturing. Samples were processed for isolation of staphylococci by following standard microbiology procedures described previously [5, 20] (and summarized below).

## Species identification and antimicrobial susceptibility testing

Following overnight culturing of samples on non-selective medium (blood agar plates), Gram-positive and catalase-positive isolates were sub-cultured again on solid Brain Heart Infusion (BHI) medium and confirmed to species level by using the Phoenix 100 Identification (ID) / Antimicrobial Susceptibility Testing (AST) Automated Microbiology System from the Becton, Dickinson and Company (Franklin Lakes, New Jersey), as described previously [23, 24]. Morphologically distinct colonies with characteristic appearance of staphylococci were selected for automated identification. Antibiotic susceptibility testing by minimum inhibitory concentrations (MICs) to a panel of 16 antibiotics (cefoxitin, penicillin, ampicillin, tetracycline, trimethoprim/sulfamethoxazole, erythromycin, chloramphenicol, gentamicin, ciprofloxacin, rifampicin, clindamycin, mupirocin [high-level], teicoplanin, linezolid, vancomycin and daptomycin) was determined with the Phoenix Automated Microbiology System as previously described [23, 24] [25]. Multidrug resistance (MDR) was defined as isolates resistant to three or more classes of antimicrobials. To validate automated species'-level identification by the Phoenix ID/AST Expert System, we randomly selected 24 isolates representing presumptively identified isolates of *S. aureus*, *S. epidermidis* and *S. haemolyticus*, and molecularly re-identified them by polymerase chain reaction (PCR) as described previously [26]. To determine the SCC*mec* types among the methicillin resistant staphylococci (MRS), SCC*mec* typing was done and interpreted as described previously [17, 22, 27]. For quality control, *S. aureus* ATCC™ 29213 and *Enterococcus faecalis* ATCC™ 29212 were included in the Phoenix ID/AST panels and molecular assays.

## Detection of virulence and AMR genes

PCR was performed to detect the virulence and AMR encoding genes, which tend to co-exist in invasive strains [15, 16]. Crude genomic DNA used as template in the PCRs was extracted by boiling for 3–5 minutes, freshly cultured cells re-suspended in 200 μl of Tris-EDTA (TE) buffer containing 80 U/ml lysostaphin. The Panton Valentine Leukocidin encoding genes (*LukS-PV* and *LukF-PV*), as well as the *ica*/*D*, *bhp*, *atlE*, *hla*, *hld*, *hlg*, *tsst* and *sea* genes, were detected by PCR as described previously [19, 28]. Similarly, presence of the genes encoding the aminoglycoside-modifying enzymes (AMEs) i.e. *aac(6′)-Ie-aph (2″)-Ia* (bifunctional aminoglycoside-6-N-acetyltransferase/2″-O-phosphoryltransferase),

*aph(3')-IIIa* (aminoglycoside-3'-O-phosphoryltransferase III) and *ant(4')-Ia* (aminoglycoside-4'-O-nucleotidyltransferase I), as well as the *vanA/vanB1* genes that encode vancomycin resistance variants, was determined by PCR as described previously [19, 28, 29].

## Results

### Nasopharyngeal colonization rates and *Staphylococcus* species identified

Staphylococci were isolated from 485 of the 764 children (63.5%, 485/764); there was no growth of staphylococci or other bacteria in samples from 279 children (36.5%, 279/764). The 485 samples with *Staphylococcus* growth yielded a total of 513 isolates belonging to 13 species, Table 1. *S. aureus* was the most prevalent species at 37.6% (193/513) followed by *S. epidermidis* (25.5%, 131/513), *S. haemolyticus* (2.3%, 12/513), *S. hominis* (0.8%, 4/513) and *S. haemolyticus/lugdunensis* (0.58%, 3/513), Table 1 and S1 Table. However, the overall prevalence of CoNS including the 156 CoNS isolates that could not be identified to species level was 42% (320/764), higher than that of *S. aureus*, Table 1. As expected the sole isolate of *S. intermedius* was coagulase-positive but *nuc*-PCR negative, and it was identified to species level by the Phoenix ID/AST expert system.

Overall, nasopharyngeal colonization rates by the dominant species was as follows: 25.3% (193/764) *S. aureus*, 17.1% (131/764) *S. epidermidis*, and 1.6% (12/764) *S. haemolyticus*. A total of 24 (4.95%, 24/485) children were co-colonized by two or three *Staphylococcus* species and the commonest co-colonizing species were *S. aureus* and *S. epidermidis*. Of note, 66.7% (16/24) of the co-colonized children harbored a multidrug resistant *Staphylococcus* while 58.3% (14/24) had a MRS; only three of the co-colonized children harbored distinct MRS species, Table 2.

### Antimicrobial susceptibility patterns

The overall prevalence of methicillin resistance was high at 23.8% (122/513); per species methicillin resistance was highest in *S. haemolyticus* (66.7%, 8/12) followed by *S. aureus*

**Table 1. *Staphylococcus* species recovered from the children (n = 764).**

| Species | Sub-total[a] n, (%) | MDR n, (%) | Methicillin resistant n, (%) | *mecA* n, (%) |
|---|---|---|---|---|
| *S. aureus* | 193/513 (38) | 58/193 (30) | 55/193 (29) | 55/193 (29) |
| *S. epidermidis* | 131/513 (26) | 14/131 (11) | 31/131 (24) | 31/131 (24) |
| *S. haemolyticus* | 12/513 (2) | 12/12 (100) | 8/12 (67) | 8/12 (67) |
| *S. hominis* | 4/513 (1) | 1/4 (25) | 0 | 0 |
| *S. haemolyticus/ lugdunensis* | 3/513 (1) | 0 | 2/3 (67) | 2/3 (67) |
| *S. pasteuri* | 3/513 (1) | 3/3 (100) | 3/3 (100) | 3/3 (100) |
| *S. lentus* | 2/513 (0.4) | 0 | 1 | 1 |
| *S. capitis* | 2/513 (0.4) | 1 | 0 | 0 |
| *S. xylosus* | 2/513 (0.4) | 2 | 2 | 2 |
| *S. kloosii* | 2/513 (0.4) | 2 | 2 | 2 |
| *S. intermedius* | 1/513 (0.2) | 0 | 0 | 0 |
| *S. caprae* | 1/513 (0.2) | 1 | 1 | 1 |
| *S. sciuri* | 1/513 (0.2) | 1 | 1 | 1 |
| Other CoNS[b] | 156/513 (30) | 9/156 (6) | 16/156 (10) | 16/156 (10) |
| **Total** | **513** | **104/513 (20)** | **122/513 (24)** | **122/513 (24)** |
| Co-colonized[c] | 24 children (5) | 17 children (71) | 13 children (54) | 13 children (54) |

[a]Of 513 staphylococcal isolates recovered.

[b]Refers to other coagulase negative staphylococci that could not be identified to species level

[c]Of 485 children in whom staphylococci were isolated

**Table 2. Co-colonization of the children by various Staphylococcus species.**

| Child # | Species | Bacteriological evaluation | | | | | | | | | | |
|---|---|---|---|---|---|---|---|---|---|---|---|---|
| | | MRS | MDR | *mecA* | *vanA* | AMEs | *LukS-PV/LukF-PV* | *tst1* | *ica* | *hla* | *hld* | *bhp* |
| H0005 | *S. aureus* | No | No | - | - | - | - | - | - | + | + | - |
| | *S. epidermidis* | No | No | - | + | - | - | - | - | + | + | + |
| H0007 | *S. aureus* | No | **Yes** | - | - | - | - | - | - | - | + | - |
| | *S. epidermidis* | No | No | - | - | - | - | - | - | - | - | - |
| **H0008** | *S. aureus* | **Yes** | **Yes** | + | - | - | - | - | - | + | + | - |
| | *S. epidermidis* | No | No | - | - | - | - | - | - | - | - | - |
| H0231 | *S. aureus* | No | **Yes** | - | - | - | - | - | - | - | - | - |
| | *S. epidermidis* | No | No | - | + | - | - | - | - | + | + | - |
| H0354 | *S. aureus* | No | No | - | - | - | - | - | - | - | - | - |
| | *S. epidermidis* | No | No | - | - | - | - | - | - | - | - | - |
| **H0184** | *S. aureus* | No | No | - | - | - | - | - | - | - | - | - |
| | *S. haemolyticus* | **Yes** | **Yes** | + | - | - | - | - | - | + | + | + |
| **H0403** | *S. aureus* | **Yes** | **Yes** | + | - | - | - | - | - | - | - | - |
| | *S. xylosus* | **Yes** | **Yes** | + | - | - | - | - | - | - | - | - |
| **H0473** | *S. aureus* | No | **Yes** | - | - | - | - | + | - | + | + | - |
| | *S. kloosii* | **Yes** | **Yes** | + | - | - | - | - | - | + | + | - |
| **H0023** | *S. aureus* | **Yes** | No | + | - | - | - | - | - | + | + | - |
| | CoNS | **Yes** | No | + | - | - | - | - | - | + | + | - |
| **H0271** | *S. aureus* | No | **Yes** | - | - | - | - | - | - | - | - | - |
| | *S. epidermidis* | **Yes** | **Yes** | + | - | - | - | - | - | - | + | - |
| | *S. haemolyticus / lugdunensis* | **Yes** | No | + | - | - | - | - | - | - | + | - |
| **H0029** | *S. aureus* | No | No | - | - | - | - | - | - | - | - | - |
| | *S. epidermidis* | **Yes** | No | + | - | + | - | - | - | + | + | - |
| | CoNS | No | No | - | - | - | - | - | - | - | - | - |
| **H0034** | *S. aureus* | **Yes** | **Yes** | + | - | - | - | - | - | + | + | - |
| | *S. epidermidis* | **Yes** | **Yes** | + | - | - | - | - | - | + | - | - |
| | *S. hominis* | No | No | - | - | - | - | - | - | - | - | - |
| **H0057** | *S. aureus* | No | No | - | - | - | - | - | - | - | - | - |
| | *S. haemolyticus* | **Yes** | **Yes** | - | - | - | - | - | - | + | + | + |
| | *S. pasteuri* | **Yes** | **Yes** | - | - | - | - | - | - | + | + | + |
| **H0108** | *S. epidermidis* | **Yes** | **Yes** | + | - | + | - | - | - | - | + | - |
| | *S. haemolyticus* | **Yes** | **Yes** | + | - | + | - | - | - | - | + | - |
| **H0020** | *S. epidermidis* | **Yes** | **Yes** | + | - | - | - | - | - | - | - | - |
| | *S. haemolyticus* | **Yes** | **Yes** | + | - | - | - | - | - | - | - | - |
| H0251 | *S. epidermidis* | No | No | - | - | - | - | - | - | - | - | - |
| | *S. haemolyticus* | No | **Yes** | - | - | - | - | - | - | - | - | - |
| H0328 | *S. epidermidis* | No | No | - | - | - | - | - | - | - | - | - |
| | *S. haemolyticus* | No | **Yes** | - | - | - | - | - | - | - | - | - |
| H0332 | *S. epidermidis* | No | No | - | - | - | - | - | - | - | - | - |
| | *S. haemolyticus* | No | **Yes** | - | - | - | - | - | - | - | - | - |
| H0185 | *S. epidermidis* | No | No | - | - | - | - | - | - | - | - | - |
| | *S. haemolyticus* | No | **Yes** | - | - | - | - | - | - | - | - | - |
| H0160 | *S. epidermidis* | No | No | - | - | - | - | - | - | - | - | - |
| | *S. haemolyticus / lugdunensis* | No | No | - | - | - | - | - | - | - | - | - |
| **H0067** | *S. epidermidis* | No | No | - | - | - | - | - | - | - | - | - |
| | *S. haemolyticus / lugdunensis* | **Yes** | No | + | - | - | - | - | - | - | - | - |

*(Continued)*

**Table 2.** (Continued)

| Child # | Species | Bacteriological evaluation | | | | | | | | | | |
|---|---|---|---|---|---|---|---|---|---|---|---|---|
| | | MRS | MDR | *mecA* | *vanA* | AMEs | *LukS-PV/LukF-PV* | *tst1* | *ica* | *hla* | *hld* | *bhp* |
| H0150 | *S. epidermidis* | No | No | - | - | - | - | - | - | - | - | + |
| | *S. intermedius* | No | No | - | - | - | - | - | - | - | - | - |
| **H0227** | *S. epidermidis* | No | No | - | - | - | - | - | - | - | - | - |
| | *S. sciuri* | **Yes** | **Yes** | + | - | - | - | - | - | - | - | - |
| **H0341** | *S. epidermidis* | **Yes** | **Yes** | + | - | - | - | - | - | - | + | - |
| | *S. capitis* | No | No | + | - | - | - | - | - | - | + | - |

Children co-colonized with a MRS are indicated in boldface font. +, PCR positive (gene detected); -, PCR negative (gene not detected).

(28.5%, 55/193) and *S. epidermidis* (23.7%, 31/131). The overall prevalence of MRCoNS was 20.9% (67/320). All MRS isolates were *mecA* positive however, they were fully susceptible to anti-MRSA agents–teicoplanin, linezolid, vancomycin and daptomycin. Also, MRS isolates were highly susceptible to mupirocin, rifampicin, chloramphenicol, and clindamycin, S1 and S2 Tables. Overall, with the exception of penicillin/methicillin, antimicrobial resistance rates were generally low especially to chloramphenicol, rifampicin, clindamycin, vancomycin, teicoplanin, linezolid and daptomycin, Table 3 and S1 Table. Five isolates with high-level mupirocin resistance (HLMup^r) were identified, four of which were methicillin susceptible *S. aureus*

**Table 3. Summary of antimicrobial susceptibility profiles of staphylococci from children in rural eastern Uganda (n = 513).**

| Species | No. isolates | Resistance to a panel of 15 antimicrobials n, (%) | | | | | | | | | | | | | | |
|---|---|---|---|---|---|---|---|---|---|---|---|---|---|---|---|---|
| | | PEN | SXT | FOX | TET | ERY | GEN | CIP | CHL | MUP | CLI | RIF | TEI | LIN | VAN | DAP |
| *S. aureus* | 193 | 174/193 (90) | 73/193 (38) | 55/193 (29) | 42/193 (22) | 37/193 (19) | 31/193 (16) | 22/193 (11) | 6/193 (3) | 4/193 (2) | 3/193 (2) | 0 | 0 | 0 | 0 | 0 |
| *S. epidermidis* | 131 | 124/131 (95) | 13/131 (10) | 31/131 (24) | 22/131 (17) | 11/131 (8) | 4/131 (3) | 4/131 (3) | 0 | 0 | 1/131 (1) | 0 | 0 | 0 | 0 | 0 |
| *S. haemolyticus* | 12 | 12/12 (100) | 12/12 (100) | 8/12 (67) | 7/12 (58) | 11/12 (92) | 2/12 (17) | 4/12 (33) | 0 | 0 | 1/12 (8) | 0 | 0 | 0 | 0 | 0 |
| *S. hominis* | 4 | 4 | 1 | 0 | 1 | 0 | 0 | 0 | 0 | 0 | 0 | 0 | 0 | 0 | 0 | 0 |
| *S. haemolyticus / lugdunensis* | 3 | 2 | 0 | 2 | 1 | 0 | 0 | 0 | 0 | 0 | 0 | 0 | 0 | 0 | 0 | 0 |
| *S. pasteuri* | 3 | 3 | 3 | 3 | 2 | 3 | 1 | 3 | 0 | 0 | 2 | 1 | 0 | 0 | 0 | 0 |
| *S. lentus* | 2 | 2 | 1 | 1 | 1 | 0 | 0 | 0 | 0 | 0 | 1 | 0 | 0 | 0 | 0 | 0 |
| *S. capitis* | 2 | 2 | 1 | 0 | 0 | 1 | 0 | 0 | 0 | 0 | 0 | 0 | 0 | 0 | 0 | 0 |
| *S. xylosus* | 2 | 2 | 2 | 2 | 2 | 2 | 0 | 2 | 0 | 0 | 2 | 0 | 0 | 0 | 0 | 0 |
| *S. kloosii* | 2 | 2 | 2 | 2 | 2 | 2 | 1 | 1 | 0 | 0 | 1 | 0 | 0 | 0 | 0 | 0 |
| *S. intermedius* | 1 | 1 | 0 | 0 | 0 | 0 | 1 | 0 | 0 | 0 | 0 | 0 | 0 | 0 | 0 | 0 |
| *S. caprae* | 1 | 1 | 1 | 1 | 1 | 1 | 1 | 0 | 0 | 0 | 0 | 0 | 0 | 0 | 0 | 0 |
| *S. sciuri* | 1 | 1 | 0 | 1 | 0 | 0 | 0 | 1 | 0 | 1 | 0 | 0 | 0 | 0 | 0 | 0 |
| Other CoNS | 156 | 134/156 (86) | 16/156 (10) | 16/156 (10) | 19/156 (12) | 11/156 (7) | 3/156 (2) | 3/156 (2) | 0 | 0 | 0 | 0 | 0 | 0 | 0 | 0 |
| **Total** | **513** | **462/513 (90)** | **127/513 (25)** | **122/513 (24)** | **98/513 (19)** | **79/513 (15)** | **41/513 (8)** | **40/513 (8)** | **4/513 (1)** | **4/513 (1)** | **11/513 (2)** | **1/513 (0.2)** | **0** | **0** | **0** | **0** |

CoNS denotes other coagulase negative staphylococci that were not identified to species level in this study

FOX, cefoxitin; PEN, penicillin; TET, tetracycline; SXT, trimethoprim-sulfamethoxazole; ERY, erythromycin; CHL, chloramphenicol; GEN, gentamicin; CIP, ciprofloxacin; RIF, rifampicin; CLI, clindamycin; MUP, Mupirocin High level; TEI, teicoplanin; LIN, linezolid; VAN, vancomycin; DAP, daptomycin

(MSSA) while one was a methicillin resistant *S. scuiri* (MRSS). Rifampicin resistance was detected in only one isolate, a methicillin resistant *S. pastueri* (MRSP). Compared to *S. epidermidis*, *S. aureus* isolates were more resistant to trimethoprim-sulfamethoxazole (SXT) (73/193, 38% vs. 13/131, 10%), erythromycin (37/193, 19% vs. 11/131, 8%), gentamicin (31/193, 16% vs. 4/131, 3%), ciprofloxacin (22/193, 11% vs. 4/131, 3%), cefoxitin (55/193, 29% vs. 31/131, 2%) and tetracycline (42/193, 22% vs. 22/131, 17%). As well, resistance among *S. aureus* isolates to cefoxitin (31/131, 24%), tetracycline (42/193, 22%), SXT (73/193, 38%), erythromycin (37/193, 19%), and ciprofloxacin (22/193, 11%) was less common than *S. haemolyticus* isolates (cefoxitin 8/12, 67% resistant; tetracycline 7/12, 58% resistant; SXT 12/12, 100% resistant; erythromycin 11/12, 92% resistant; ciprofloxacin 4/12, resistant), Table 3 and S1 Table. While we found few isolates of *S. pastueri*, *S. xylosus* and *S. kloosii*, they were all MRS and multidrug resistant, Table 1.

The overall prevalence of MDR was 20.3% (104/513), and 59% (72/122) of the MRS were multidrug resistant, S2 Table. All *S. haemolyticus* and *S. pastueri* isolates were multidrug resistant while 30.1% (58.1/513) and 10.8% (14/131) of *S. aureus* and *S. epidermidis* isolates respectively, were multidrug resistant, Table 1. The prevalence of MDR was also high among the methicillin resistant *S. aureus* (MRSA) and methicillin resistant *S. epidermidis* (MRSE) i.e. 64.2% (34/53) and 42.4% (14/33), respectively. As expected, the methicillin resistant isolates were generally resistant to the non-beta lactam antimicrobials i.e. tetracycline (53%), SXT (55%), erythromycin (42%), and ciprofloxacin (32%). Overall, only 14.4% of CoNS were multidrug resistant but the MDR phenotype was generally high among MRCoNS.

The most frequent SCC*mec* type among MRS was type I (32.8%, 40/122) while types II and III were detected in 8.2% (10/122) and 6.6% (8/122) of MRS, respectively. SCC*mec* types I+II+III combined occurred in 47.5% (58/122) of MRS isolates. On the other hand, SCC*mec* types IV and V were detected in 18% (22/122) and 3.3% (4/122) of MRS, respectively, while SCC*mec* types IV+V combined occurred in 21.3% (29/122) isolates. Among MRSA, SCC*mec* types I and IV were the most frequent at 38.2% (21/55) and 31% (17/55) respectively; seven isolates were not type-able (NT). Likewise among MR-CoNS, SCC*mec* type I was the most prevalent at 6% (19/320) followed by types II (2.2%, 7/320), III (2.2%, 7/320) and IV (2%, 6/320); 28 MRCoNS were not type-able.

## Virulence and antimicrobial resistance genes

In this study, all isolates were investigated for carriage of AMR and virulence genes in addition to the *mecA* gene described above. The most frequent AMR genes in *S. aureus* isolates were *vanA*, *ant(4′)-Ia*, and *aac(6′)-Ie- aph(2″)-Ia* while the most frequent virulence genes were *hld* and *hla*, Table 4. Although phenotypic resistance to vancomycin was not detected, 20 isolates carried the *vanA* gene while 6 carried the *vanB* gene. Eleven (42%, 11/26) of the isolates with *vanA/vanB* also carried the *mecA* gene. Furthermore, 62 isolates carried the AMEs genes (any of *aac(6′)-Ie-aph(2″)-Ia*, *ant(4′)-Ia* and *aph(3′)-IIIa*), of which five (8%) were phenotypically resistant to gentamicin (an aminoglycoside). Generally, the *hld*, *hla*, *sea* and *LukS-PV / LukF-PV* genes were more frequent in *S. aureus* compared to CoNS species, Table 4. Additionally, in *S. aureus* the *bhp* gene was more frequent in MRSA isolates compared to MSSA while in CoNS, the *sea*, *sdrE*, *bhp*, *hla*, and *hld* genes were more frequent in MRSCoNS, Table 4. Among the CoNS, the AMR genes were more frequent in *S. epidermidis* compared to other species. The *hlg* gene was detected in only four isolates while *hlb* and *ica/D* genes were not detected. Overall, the virulence and AMR genes were generally more frequent in *S. aureus* compared to CoNS species (for example, the *LukS-PV / LukF-PV* genes were only detected in *S. aureus*).

**Table 4. Frequency of virulence and antimicrobial resistance genes in staphylococci.**

| Gene | Function | No. isolates positive (%) | | | | | |
|---|---|---|---|---|---|---|---|
| | | **S. aureus** | | | **CoNS** | | |
| | | **All S. aureus** | **MRSA** | **MSSA** | **All CoNS** | **MRS** | **MSS** |
| **Antimicrobial resistance genes** | | | | | | | |
| mecA | Methicillin resistance | 55/193 (29) | 55/55 (100) | 0 | 67/320 (21) | 67/67 (100) | 0 |
| vanA | Vancomycin resistance (high-level) | 11/193 (6) | 4/55 (7) | 7/138 (5) | 9/320 (3) | 3/67 (5) | 6/253 (2) |
| vanB | Vancomycin resistance (low-level) | 5/193 (3) | 5/55 (9) | 2/138 (1) | 1/320 (0.3) | 1/67 (2) | 0 |
| aac(6')-Ie- aph(2'')-Ia | AME* (GEN-R, TOB-R, AMK-R, KAN-R) | 9/193 (5) | 2/55 (4) | 7/138 (5) | 11/320 (3) | 8/67 (12) | 3/253 (1) |
| ant(4')-Ia | AME* (GEN-S, TOB-R, AMK-R, KAN-R) | 14/193 (7) | 3/55 (6) | 11/138 (8) | 10/320 (3) | 7/67 (10) | 3/253 (1) |
| aph(3')-IIIa | AME* (GEN-S, TOB-S, AMK-R, KAN-R) | 3/193 (2) | 0 | 3/138 (2) | 3/320 (1) | 3/67 (5) | 0 |
| **Virulence genes** | | | | | | | |
| LukS-PV / LukF-PV** | Bicomponent leukocidin; pore-forming toxin; kills leukocytes | 23/193 (12) | 10/55 (18) | 13/138 (9) | 0 | | |
| sea | Staphylococcal enterotoxin A | 18/193 (9) | 3 (6) | 15/138 (11) | 15/320 (5) | 12/67 (18) | 3/253 (1) |
| tst1 | Endothelial toxicity (direct & cytokine-mediated); super-antigen activity | 4/193 (2) | 3 (6) | 1/138 (1) | 2/320 (1) | 2/67 (3) | 0 |
| sdrE | Adhesin | 10/193 (5) | 3 (6) | 7/138 (5) | 13/320 (4) | 11/67 (16) | 2/253 (1) |
| bhp | Cell wall associated biofilm protein | 22/193 (11) | 11 (20) | 11/138 (8) | 28/320 (9) | 16/67 (24) | 12/253 (5) |
| hla | Cytolytic pore-forming toxin | 54/193 (28) | 19 (35) | 35/138 (25) | 48/320 (15) | 32/67 (48) | 16/253 (6) |
| hld | Cytolytic toxin; binds neutrophils & monocytes | 77/193 (40) | 24 (44) | 53/138 (38) | 85/320 (27) | 53/67 (79) | 32/253 (13) |
| Hlg | Bicomponent leukocidin; hemolysis | 1/193 (1) | 1 (2) | 0 | 3/320 (1) | 2/67 (3) | 1/253 (0.4) |

*AME(s) denotes aminoglycoside-modifying enzyme(s) AAC(6')/APH (2"), ANT(4')-I and APH(3')-III encoded by the following genes: aac(6')-Ie- aph(2'')-Ia, ant(4')-Ia and aph(3')-IIIa, respectively; GEN, gentamicin; TOB, tobramycin; AMK, amikacin; KAN, kanamycin; S, susceptible, R, resistant.

**LukS-PV / LukF-PV, panton-valentine leukocidin; sea; tst1, toxic shock syndrome toxin-1; sdrE, Serine-aspartate repeat protein E; bhp, biofilm associated protein; hla, alpha-toxin; hlb, beta-toxin; hld, delta-toxin; hlg, gamma-toxin

## Discussion

In this study, we used a rigorous approach to determine the species and characteristics of staphylococci colonizing children in Eastern Uganda. The Automated system we used i.e. the Phoenix AST/ID Expert System, is more accurate than the manual methods at identifying Gram-positive bacteria, as well as AMR testing [30]. However, in the developing countries, manual methods are the mainstay for identification of staphylococci, and in most cases the tube coagulase test is the confirmatory test for identification of *S. aureus* [5]. The tube coagulase test is considered to be more reliable at identifying *S. aureus* when a firm clot that doesn't move on tilting the tube occurs; therefore, the subjectivity in interpreting the tube coagulase test probably leads to misidentification of CoNS as *S. aureus* [31]

In this study, a high staphylococcal nasopharyngeal colonization rate (63.5%, 485/764) was reported for healthy children in Eastern Uganda. However, our rate is comparable to the rates reported by other investigators e.g. ≥80% by Budri et al [32] and Faria et al [1]. Furthermore, the number of Staphylococcus species (i.e. 13) in this study is also comparable to the 9 species reported by Faria et al in Denmark [1], but less than the 19 species reported by Xu et al in a study of staphylococcal colonization in healthy individuals and the environment in London

[14]. *S. aureus* was the most prevalent species followed by *S. epidermidis*, which contradicts reports of *S. epidermidis* as the most prevalent Staphylococcus species among staphylococcal isolates from humans [1, 14, 32]. However, as the overall prevalence of CoNS in this study (i.e. 42%, 320/764) was higher than that of *S. aureus* (37.6%, 193/513), the prevalence of *S. epidermidis* might be higher than what we have reported had we succeeded in identifying all the CoNS to species level. Still, several factors could have affected our recovery of *S. epidermidis*; for instance, unlike *S. aureus*, CoNS especially *S. epidermidis* mainly colonize the skin [33] yet we sampled the nasopharynx which is mainly colonized by *S. aureus*. Additionally, the enrichment methods employed in isolation of staphylococci generally favor recovery of *S. aureus* [32]. Interestingly, while majority of the CoNS species identified are human-associated (i.e. the *S. epidermidis*-like group–*S. epidermidis*, *S. haemolyticus*, *S. capitis*, *S. hominis*, etc. [2]), animal-associated CoNS species (i.e. *S. caprae*) and animal-associated CoPS species (i.e. *S. intermedius*) were also detected, which perhaps reflects a rural, agrarian population typical of Eastern Uganda. The detection of animal-associated staphylococci in this setting is a cause for concern as it may reflect (1) contamination of human samples by animal-associated strains, (2) occurrence of animal-associated CoPS in human samples, which could yield false-positive results on coagulase testing and overestimation of *S. aureus*/MRSA rates [30].

Around 5% of the children in this study were colonized by two or three Staphylococcus species, fewer than reports by investigators from Europe (i.e. 7.5% to 30%) [1, 3, 32]. In agreement with previous reports [1, 3, 32], the commonest co-colonizing species were *S. aureus* and *S. epidermidis*. Only one of the co-colonized children harbored *S. aureus* and *S. haemolyticus/lugdunensis*, which is not surprising as *S. lugdnensis* produces lugdunin, an antibacterial agent that reduces the probability of *S. aureus* colonization [3, 34]. Further, 66.7% (16/24) of the co-colonized children harbored a multidrug resistant Staphylococcus while 58.3% (14/24) harbored a MRS. However, only three of the co-colonized children harbored different species of MRS, and this is in agreement with the previously reported negative correlation of co-colonization by distinct MRS species [1, 3, 32].

Antimicrobial resistance rates were generally low across species. However, in sharp contrast to reports from the Nordic countries [35], penicillin resistance and MRS nasopharyngeal colonization rates were quiet high, which is consistent with global reports of increasing MRS prevalence in the community [36]. The detection of SCC*mec* types I+II+III combined and SCC*mec* types IV+V combined suggests coexistence of hospital-associated and community-associated MRS that seems to result from dissemination of hospital-associated strains into the community [36]. Further, there were five isolates with high-level mupirocin resistance (HLMup$^r$), four of which were MSSA while one was a methicillin resistant *S. scuiri* (MRSS). This is consistent with increasing levels of mupirocin resistance staphylococci, especially in MRCoNS [3] [37], which is worrying as mupirocin is the drug of choice for eradication of Staphylococcus colonization. The occurrence of HLMup$^r$ in Africa before widespread use of mupirocin for MRSA decolonization requires continuous monitoring.

With the exception of *mecA*-positive isolates, majority of the isolates harboring AMR genes i.e. *vanA/vanB* and AMEs (any of *aac(6′)-Ie-aph(2″)-Ia*, *ant(4′)-Ia* and *aph(3′)-IIIa*) were phenotypically susceptible to drug(s) targeted by the gene products. For example, phenotypic resistance to vancomycin was not detected; however, we found 26 isolates with the *vanA/vanB* genes, and 42% (11/26) of them carried *mecA*. *vanA* and *vanB* genes encode high- and low-level resistance to vancomycin, respectively. *vanA* type of resistance is widely spread in enterococci and it is transferrable between enterococci and staphylococci [29, 38, 39]. While phenotypic resistance to vancomycin among enterococci and staphylococci is rare in Uganda [24, 40], *vanA/vanB* PCR-positive vancomycin-susceptible isolates of enterococci and staphylococci have been reported in Uganda [28, 40]. While it is puzzling, *vanA*-positive-vancomycin-

susceptible enterococci do occur, and they have also been reported from the developed countries [39, 41, 42]. These isolates have been found to lack key genes e.g. *vanR* and *vanS*, which are required for activation of transcription of the vancomycin resistance genes. Additionally, the *vanA*-positive-vancomycin-susceptible isolates can possess insertion sequences e.g. the *IS*L3-family element that silence transcription of the *vanA* operon [39, 41, 42]. The fact that horizontally transferrable silenced-*vanA* successfully reverted into resistance during vancomycin treatment [39], genotypic testing of invasive vancomycin-susceptible isolates has been recommended [33]. Relatedly, while phenotypic resistance to aminoglycosides correlates well with the detection of AMEs genes [43, 44], in this study, few isolates with AMEs genes (i.e. 8%, 5/62) were phenotypically resistant to gentamicin (an aminoglycoside). However, gentamicin-susceptible isolates positive for AMEs genes have been reported before [37, 38]. Overall, the carriage of AMR genes in absence of significant antibiotic selective pressure points to presence of AMR drivers in the community that should be investigated [32, 36] with robust approaches like whole genome sequencing [42] to better understand the mechanisms underlying altered AMR susceptibility in our setting.

In this study, the virulence and AMR genes were generally more frequent among *S. aureus* isolates compared to CoNS species. For example, the *LukS-PV / LukF-PV* genes were detected only in *S. aureus* but not CoNS. The *hlg* gene was detected in only four *S. aureus* isolates, while *hlb* and *ica/D* were not detected at all. The low frequency of virulence genes in CoNS is consistent with reports that CoNS generally possess a smaller repertoire of virulence genes compared to *S. aureus* [2], and that human-associated *S. aureus* strains do not express beta toxins (*hlb*) [33]. Furthermore, the *bhp* gene was prevalent in *S. aureus* and it was significantly associated with MRSA. Interestingly, *bhp* encodes a protein homologue of Bap (biofilm associated protein) and it is generally absent in human-associated *S. aureus* strains. However, the Bap homologue termed bhp, is present in *S. epidermidis* [2]. Investigators in the developed countries found an association between presence of *bhp* and *aacA [aac(6)-aph(2)]* genes with the "not cured" clinical outcome among patients who underwent surgery and got infected by staphylococci [45].

The fact that staphylococci are opportunistic pathogens, identification of virulence/invasive strains is critically important in guiding diagnosis and treatment of staphylococcal infections [46]. To this end, the detection of staphylococcal virulence factors, especially genes involved in biofilm formation, is frequently cited as a means through which we can rapidly identify invasive strains [19, 28, 46]. Compared to previously characterized staphylococcal isolates in Uganda [19, 28], we noted that the virulence and AMR genes are significantly more prevalent in *S. aureus* and generally in hospital isolates. In fact, the detection of *icaA/D* and *hlg* genes have been reported in hospital isolates but not community isolates [19, 28], and current study conforms to this notion.

The polysaccharide intercellular adhesin/poly-N-acetylglucosamine (PIA/PNAG) is associated with biofilm formation in staphylococci especially *S. epidermidis*. PIA/PNAG is encoded by the *ica* gene cluster, which comprises the *icaA*, *icaD*, *icaB* and *icaR* genes. Deletion of these genes is associated with biofilm absence [2, 46]. Biofilm production and presence of the *ica* operon correlate with disease causing clinical isolates and in mouse models, the PIA/PNAG-negative mutants were significantly less pathogenic [2, 28]. The *bhp/Bap* genes are also associated with the biofilm phenotype and may be useful as markers in distinguishing between pathogenic strains and normal flora [46] [2]. Overall, the implication of detecting these genes in flora of children at the IMHDSS, albeit at lower frequencies relative to hospital-associated isolates, is that (a) strains bearing these genes could be potentially virulent and, (b) the genes could be useful as markers for screening potentially invasive strains.

## Conclusions

The staphylococcal nasopharyngeal colonization rate in children in Eastern Uganda is high but comparable to rates from other countries. Also, the Staphylococcus species' distribution in the children mirrors what has been described from other settings but with *S. aureus* as the dominant species. Approximately 5% of the children were colonized by two or three Staphylococcus species but co-colonization by distinct MRS species is rare. Although the AMR rates were low, nasopharyngeal colonization by MRS is quiet high. The detection of high-level mupirocin resistance requires further investigation. Furthermore, while majority of CoNS and CoPS were human-associated, the detection of animal-associated CoNS and CoPS implies that individuals at the IMHDSS could be infected by zoonotic staphylococci. Importantly, with the occurrence of animal-associated CoPS species in human samples, tube-coagulase testing probably overestimates *S. aureus*/MRSA rates in this setting. Therefore, species-level identification of staphylococci with robust approaches is advised, especially clinically relevant isolates. The carriage of AMR genes in community-associated isolates, especially the *vanA*-positive-vancomycin-susceptible bacteria, points to existence of AMR drivers in the community that should be investigated for a better understanding of genetic determinants of altered antibiotic susceptibility. Overall, virulence and AMR genes, especially *ica* and *bhp*/*Bap*, were significantly more prevalent in *S. aureus*, could be useful as markers for screening potentially invasive strains.

## Supporting information

**S1 Table. General characteristics of community-associated staphylococci isolated from children less than 5 years in rural Eastern Uganda.**
(XLS)

**S2 Table. Details of drug resistance profiles of methicillin resistant staphylococci from children (n = 122).**
(DOCX)

## Acknowledgments

We thank the parents/guardians of the children for accepting to participate in the study. We also thank the staff and management of the Clinical Microbiology and Molecular Biology Laboratories of Makerere University College of Health Sciences for technical assistance.

## Author Contributions

**Conceptualization:** David Patrick Kateete.

**Data curation:** David Patrick Kateete, Benon B. Asiimwe, Christine Florence Najjuka.

**Formal analysis:** David Patrick Kateete, Benon B. Asiimwe, Christine Florence Najjuka.

**Investigation:** David Patrick Kateete, Christine Florence Najjuka, Elizeus Rutebemberwa.

**Methodology:** David Patrick Kateete, Benon B. Asiimwe, Raymond Mayanja, Christine Florence Najjuka, Elizeus Rutebemberwa.

**Project administration:** David Patrick Kateete.

**Resources:** Elizeus Rutebemberwa.

**Supervision:** David Patrick Kateete, Christine Florence Najjuka.

**Validation:** David Patrick Kateete, Christine Florence Najjuka.

**Visualization:** David Patrick Kateete.

**Writing – original draft:** David Patrick Kateete, Benon B. Asiimwe, Elizeus Rutebemberwa.

**Writing – review & editing:** David Patrick Kateete.

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
