## [Decision Letter · Decision Letter 0]

8 Nov 2019

PONE-D-19-21368

Species and susceptibility profiles of staphylococci isolated from healthy children in Eastern Uganda

PLOS ONE

Dear Dr. Kateete,

Thank you for submitting your manuscript to PLOS ONE. After careful consideration, we feel that it has merit but does not fully meet PLOS ONE’s publication criteria as it currently stands. Therefore, we invite you to submit a revised version of the manuscript that addresses the points raised during the review process.

I have received the reviews of your manuscript. While your paper addresses an interesting question, the reviewers stated several concerns about your study that need to be addressed.  Please see reviewers' insightful comments below.  Personally, I also have a few concerns, please see specific comments below.  In addition, there are quite a few awkward sentences throughout the manuscript, please thoroughly copyedit your manuscript for language usage, spelling, and grammar.

Specific comments:

1. The rational in abstract should be strengthened.

2. Line 44 and 47, please use Staphylococcus aureus on line 44 and use S. aureus on line 47.

3. Line 56, please spell out SCCmec since this is the first time it was mentioned.

We would appreciate receiving your revised manuscript by Dec 23 2019 11:59PM. To enhance the reproducibility of your results, we recommend that if applicable you deposit your laboratory protocols in protocols.io, where a protocol can be assigned its own identifier (DOI) such that it can be cited independently in the future. For instructions see: http://journals.plos.org/plosone/s/submission-guidelines#loc-laboratory-protocols

We look forward to receiving your revised manuscript.

Kind regards,

Baochuan Lin, Ph.D.

Academic Editor

PLOS ONE

Journal Requirements:

2. Please provide additional details regarding participant consent. In the ethics statement in the Methods and online submission information, please ensure that you have specified (1) whether consent was suitably informed and (2) what type you obtained (for instance, written or verbal). Since your study included minors under age 18, state whether you obtained consent from parents or guardians. If the need for consent was waived by the ethics committee, please include this information.

Reviewers' comments:

Reviewer's Responses to Questions

**Comments to the Author**

1. Is the manuscript technically sound, and do the data support the conclusions?

Reviewer #1: Yes

Reviewer #2: Yes

2. Has the statistical analysis been performed appropriately and rigorously? 

Reviewer #1: Yes

Reviewer #2: No

3. Have the authors made all data underlying the findings in their manuscript fully available?

Reviewer #1: Yes

Reviewer #2: Yes

4. Is the manuscript presented in an intelligible fashion and written in standard English?

Reviewer #1: Yes

Reviewer #2: Yes

5. Review Comments to the Author

Reviewer #1: In this work, Kateete and colleagues determined the species and the antibiotic susceptibility of Staphylococci isolated from 764 healthy children under 5 years of age residing at the Iganga/Mayuge Health & Demographic Surveillance Site (IMHDSS) in Eastern Uganda. They isolated 513 staphylococci strains belonging to 13 different species. The most prevalent was Staphyloccoccus aureus, followed by S. epidermidis (25,5%), S. heamolyticus (2,3%), S. hominis (0,8%), and S. haemolyticus/lugduniensis (0.58%). The staphylococcal carriage rates from healthy children in rural Eastern Uganda were comparable to those from other countries showing a high level of virulence and AMR genes particularly in S. aureus strains.

Although restricted to a specific area in Africa, the identification of virulent staphylococci may offer some relevant information to the local clinical community, especially with the widespread occurrence of multidrug-resistant organisms and the increasing number of immunocompromised persons in the population. Continuous epidemiological surveillance would also contribute to reducing the Staphylococcal infection burden, hospital stay, and infection morbidity.

The manuscript is a well written, organized, and thorough presentation of the topic. I enjoyed reading the article and found it both relevant in its possible implications for the prevention and the development of targeted treatments and preventive strategies against invasive strains in children.

There are some minor typos throughout the text, which can be easily corrected.

Reviewer #2: This study reports the prevalence of staphylococcal nasopharyngeal colonisation amongst healthy children in Eastern Uganda. 764 children were included, and the authors identified the species of each isolate, their antimicrobial susceptibility and also tested for some virulence and resistance genes.

The sampling appears to have been at a single time point, so “colonisation” should be used rather than “carriage” – which suggests sampling longitudinally. Secondly, only the nasal pharynx was sampled – so “colonisation” should be “nasopharyngeal colonisation” throughout.

Acronyms should be defined when they are first used. MDR defined line 170 – but used prior. A definition of MDR needs to be provided in the methods.

Proportions have been provided to 1, 2, and 0 decimal places. This needs to be consistent, and I would recommend using whole numbers only. P values should be reported as <0.05 (and or <0.001). There is no section to explain what tests were used to derive P values – nor what hypotheses were actually being tested. In some instances I don’t think the hypothesis testing is useful – e.g. it would be more useful to the reader to list the % of each species resistant to each antibiotic than provide the low P values, which is less informative. The statistical analysis needs to be described in the methods.

How did the detection of resistance genes relate to the AST?

The genes listed should be consistent – LukSF or PVL is used (LukSF is more correct).

From line 213 the isolates are compared to another set of isolates. There is no introduction as to where these isolates came from, apart from saying that they were from a hospital. Unless a lot of further information is provided about the provenance of these isolates and why comparison with the study isolates is important or interesting then this section should be removed.

In line 126 the genus name does not require italics.

Line 143 “MRS staphylococcus” is not correct

MRS not defined until line 157.

Line 276/277 virulence genes and resistance genes confused

Line 280 does not with the preceding sentence.

6. PLOS authors have the option to publish the peer review history of their article (what does this mean?). If published, this will include your full peer review and any attached files.

Reviewer #1: No

Reviewer #2: Yes: S R Ritchie

---

## [Author Response · Author response to Decision Letter 0]

26 Nov 2019

November 26, 2019

The Academic Editor, PLOS ONE 

Dear Dr. Baochuan Lin, Ph.D.

RE: Submission of revised “PONE-D-19-21368”: Species and drug susceptibility profiles of staphylococci isolated from healthy children in Eastern Uganda.

We thank you for handling our manuscript through the peer-review process, and finding it suitable for publication in PLOS ONE. 

We carefully considered and fully effected all the insightful comments from the peer-reviewers, as well as your specific concerns. We removed the awkward sentences in the manuscript. To the best of our ability, we have copy-edited the manuscript for language usage, spelling, and grammar. We hope that you will find our manuscript suitable for publication in your esteemed journal.

Again, we thank you and look forward to hearing from you soon concerning an editorial decision.

Yours sincerely,

David Patrick Kateete Ph.D.

Makerere University College of Health Sciences

Kampala, Uganda

POINT-BY-POINT RESPONSE TO THE REVIEWERS’ COMMENTS 

SPECIFIC EDITORIAL COMMENTS:

1. The rational in abstract should be strengthened.

Response:

The rationale in abstract has been strengthened, see lines 19 to 28 

2. Line 44 and 47, please use Staphylococcus aureus on line 44 and use S. aureus on line 47.

Response:

As advised, we have used Staphylococcus aureus and used S. aureus thereafter, lines 21 and 23.

3. Line 56, please spell out SCCmec since this is the first time it was mentioned.

Response:

As advised, we have spelled out SCCmec at first time of use, see line 75/76.

Journal Requirements:

2. Please provide additional details regarding participant consent. In the ethics statement in the Methods and online submission information, please ensure that you have specified (1) whether consent was suitably informed and (2) what type you obtained (for instance, written or verbal). Since your study included minors under age 18, state whether you obtained consent from parents or guardians. If the need for consent was waived by the ethics committee, please include this information.

Response:

We affirm that our manuscript meets PLOS ONE's style requirements stipulated above, including those for file naming. We have also provided additional details regarding participant consent, see lines 96 to 104. 

REVIEWERS' COMMENTS

REVIEWER #1:

Comment(s):

In this work, Kateete and colleagues determined the species and the antibiotic susceptibility of Staphylococci isolated from 764 healthy children under 5 years of age residing at the Iganga/Mayuge Health & Demographic Surveillance Site (IMHDSS) in Eastern Uganda. They isolated 513 staphylococci strains belonging to 13 different species. The most prevalent was Staphyloccoccus aureus, followed by S. epidermidis (25,5%), S. heamolyticus (2,3%), S. hominis (0,8%), and S. haemolyticus/lugduniensis (0.58%). The staphylococcal carriage rates from healthy children in rural Eastern Uganda were comparable to those from other countries showing a high level of virulence and AMR genes particularly in S. aureus strains.

Although restricted to a specific area in Africa, the identification of virulent staphylococci may offer some relevant information to the local clinical community, especially with the widespread occurrence of multidrug-resistant organisms and the increasing number of immunocompromised persons in the population. Continuous epidemiological surveillance would also contribute to reducing the Staphylococcal infection burden, hospital stay, and infection morbidity.

The manuscript is a well written, organized, and thorough presentation of the topic. I enjoyed reading the article and found it both relevant in its possible implications for the prevention and the development of targeted treatments and preventive strategies against invasive strains in children. There are some minor typos throughout the text, which can be easily corrected.

Response:

Thank you for the good remarks on our manuscript. We are delighted to hear that you enjoyed reading the manuscript, and that you found it informative regarding the widespread occurrence of antimicrobial resistance in context of immunocompromised individuals. 

As advised, we have reworked the manuscript and eliminated the typos. The revised manuscript reads much better now.

REVIEWER #2: 

Comment:

This study reports the prevalence of staphylococcal nasopharyngeal colonisation amongst healthy children in Eastern Uganda. 764 children were included, and the authors identified the species of each isolate, their antimicrobial susceptibility and also tested for some virulence and resistance genes.

The sampling appears to have been at a single time point, so “colonisation” should be used rather than “carriage” – which suggests sampling longitudinally. Secondly, only the nasal pharynx was sampled – so “colonisation” should be “nasopharyngeal colonisation” throughout.

Response:

We thank you for this and all your invaluable comments. As advised, we have now used the term “nasopharyngeal colonisation” throughout the text.

Comment:

Acronyms should be defined when they are first used. MDR defined line 170 – but used prior. A definition of MDR needs to be provided in the methods.

Response:

All acronyms are now defined at first use. A definition of multidrug resistance (MDR) has been provided in the methods (lines 130 to 131). We thank you.

Comment:

Proportions have been provided to 1, 2, and 0 decimal places. This needs to be consistent, and I would recommend using whole numbers only. P values should be reported as <0.05 (and or <0.001). There is no section to explain what tests were used to derive P values – nor what hypotheses were actually being tested. In some instances I don’t think the hypothesis testing is useful – e.g. it would be more useful to the reader to list the % of each species resistant to each antibiotic than provide the low P values, which is less informative. The statistical analysis needs to be described in the methods.

Response:

As advised, in the revised manuscript proportions have been provided consistently. Again, as you advised, we have used whole numbers only, and p-values are reported as suggested. Statistical analysis has been provided in the Methods section (lines 158-162), and it includes the specific tests we used to derive the p-values. We have used hypothesis testing only where we thought it would be useful. In Table 4 (pages 15/16) , we listed the percentages (%) in brackets for each species or isolates that was/were resistant or carrying a gene to each antibiotic. We thank you.

Comment:

How did the detection of resistance genes relate to the AST?

The genes listed should be consistent – LukSF or PVL is used (LukSF is more correct).

Response:

The genes investigated are now listed are now consistently used. 

As you advised, we have used LukSF instead of PVL. 

Regarding detection of resistance genes, apart from the mecA gene that correlated well with phenotypic methicillin/penicillin resistance, the detection of antibiotic resistance genes generally correlated poorly with antibiotic resistance. For example, of the 62 isolates that harbored the genes encoding the aminoglycoside-modifying enzymes (AMEs) (i.e. any of the aac(6')-Ie-aph(2'')-Ia, ant(4')-Ia and aph(3')-IIIa genes), only five (8%) were phenotypically resistant to gentamicin (an aminoglycoside). Additionally, although the vanA gene was detected in 20 isolates, phenotypic vancomycin resistance was not detected. Please note that such findings/observations have been reported before i.e. the detection of resistance genes may not always correlate with phenotypic resistance. We have added these additional aspects to the Results section and discussed the potential their implications in the Discussion section (see lines 241-256; 349-374). We thank you.

Comment:

From line 213 the isolates are compared to another set of isolates. There is no introduction as to where these isolates came from, apart from saying that they were from a hospital. Unless a lot of further information is provided about the provenance of these isolates and why comparison with the study isolates is important or interesting then this section should be removed.

Response:

Many thanks for raising this concern. 

The reason why we compared isolates from the IMHDSS (current study) with previously characterized isolates from the hospital was an attempt to identify key differences in the distribution of virulence and antibiotic resistance genes, which could be used to distinguish between invasive/virulent and non-invasive staphylococci. We have clarified this, and we reworked the manuscript to this effect by providing additional information (see lines 82-93; 141-162; and 376-415). The is now an introduction as to where the isolates came from (lines 264-267 and earlier in the methods, page 7). This analysis was insightful as it showed that in Uganda, virulence and resistance genes are more prevent in hospital-associated isolates especially S. aureus at Mulago hospital compared to isolates from the community.

Comment:

In line 126 the genus name does not require italics.

Response: 

The genus name has been un-italicized here, and throughout the manuscript. Thank you.

Comment:

Line 143 “MRS staphylococcus” is not correct

Response:

We agree. The phrase “MRS staphylococcus” is not correct and it has been revised, see lines 184-186. Thank you.

Comment:

MRS not defined until line 157.

Response:

This was an oversight. MRS is now defined in the Methods section where it is first used, see line 130.

Comment:

Line 276/277 virulence genes and resistance genes confused

Response:

Many thanks for pointing out this error. We have now separated virulence from resistance genes, please see lines 376 to 387. 

Comment:

Line 280 does not with the preceding sentence.

Response: 

This section has been clarified and it is now concordant, see lines 377-380. Thank you

---

## [Decision Letter · Decision Letter 1]

18 Dec 2019

PONE-D-19-21368R1

Species and drug susceptibility profiles of staphylococci isolated from healthy children in Eastern Uganda

PLOS ONE

Dear Dr. Kateete,

Thank you for submitting your manuscript to PLOS ONE. After careful consideration, we feel that it has merit but does not fully meet PLOS ONE’s publication criteria as it currently stands. Therefore, we invite you to submit a revised version of the manuscript that addresses the points raised during the review process.

Both reviewers agreed that the revised manuscript show significant improvement, however, there are still some issues/concerns that need to be addressed.  Please review and address all comments from reviewer #2.

We would appreciate receiving your revised manuscript by Feb 01 2020 11:59PM. To enhance the reproducibility of your results, we recommend that if applicable you deposit your laboratory protocols in protocols.io, where a protocol can be assigned its own identifier (DOI) such that it can be cited independently in the future. For instructions see: http://journals.plos.org/plosone/s/submission-guidelines#loc-laboratory-protocols

We look forward to receiving your revised manuscript.

Kind regards,

Baochuan Lin, Ph.D.

Academic Editor

PLOS ONE

Reviewers' comments:

Reviewer's Responses to Questions

**Comments to the Author**

1. If the authors have adequately addressed your comments raised in a previous round of review and you feel that this manuscript is now acceptable for publication, you may indicate that here to bypass the “Comments to the Author” section, enter your conflict of interest statement in the “Confidential to Editor” section, and submit your "Accept" recommendation.

Reviewer #1: All comments have been addressed

Reviewer #2: (No Response)

2. Is the manuscript technically sound, and do the data support the conclusions?

Reviewer #1: Yes

Reviewer #2: Yes

3. Has the statistical analysis been performed appropriately and rigorously? 

Reviewer #1: Yes

Reviewer #2: I Don't Know

4. Have the authors made all data underlying the findings in their manuscript fully available?

Reviewer #1: Yes

Reviewer #2: No

5. Is the manuscript presented in an intelligible fashion and written in standard English?

Reviewer #1: Yes

Reviewer #2: Yes

6. Review Comments to the Author

Reviewer #1: The authors have addressed all my suggestions. I found their responses quite satisfactory and the revised version has been much improved. I now recommend the paper for publication with no revisions.

Reviewer #2: The addition of some extra information about the hospital strains was helpful – but it is still inadequate. The methods needs to include information of how these were obtained, who from, what diseases they caused etc. If they’re from adults then careful explanation needs to be made to explain why a study of colonising strains from children would be compared to strains from a hospital obtained from adults. The two strain collections don’t really add up – all staphs vs. S. aureus and S. epi only. (My personal opinion is that this comparison actually doesn’t add a great deal to your manuscript).

Line 57 … S. aureus strains are overwhelmingly coagulase-positive, some are atypical in that they do not produce coagulase.

This requires a reference to be added – ideally several references.

Line 91 … “with the assumption that hospital-associated isolates are more invasive/virulent and/or resistant to antibiotics compared to community-associated isolates.”

Why did you have this assumption? – prior research has shown that most hospital acquired S. aureus infections are endogenous – caused by the carriage strain. This might best be stated as the null hypothesis.

Table 4 still contains P values with a lot of decimal points, as does the text. Use 1 d.p. or use <0.05; <0.001. “S. haemolyticus was significantly more resistant to cefoxitin (p=0.0055), tetracycline (p=0.0041), SXT (p<0.0001), erythromycin (p<0.0001)…”

should become:

“S. haemolyticus was significantly more resistant to cefoxitin (p<0.05), tetracycline (p<0.05), SXT (p<0.001), erythromycin (p<0.001)”

These P values are low – but they don’t help the reader understand the message. It would be better to provide the proportions. These are NOT provided in table 1.

So the text would read:

“Resistance among S. aureus isolates to cefoxitin (x/y, z%), tetracycline (x/y, z%)…..was less common than S. haem isolates (cefoxitin a/b, c% resistant; tetracycline….).

This type of information would be better in a table. Again – I don’t think the chi-sq testing is useful.

7. PLOS authors have the option to publish the peer review history of their article (what does this mean?). If published, this will include your full peer review and any attached files.

Reviewer #1: No

Reviewer #2: Yes: S R Ritchie

---

## [Author Response · Author response to Decision Letter 1]

12 Jan 2020

January 12, 2020

The Academic Editor, PLOS ONE

Dear Dr. Baochuan Lin, Ph.D.

RE: Submission of revised PONE-D-19-21368R1: Species and drug susceptibility profiles of staphylococci isolated from healthy children in Eastern Uganda

We thank you again for handling our manuscript through the peer-review process, and finding it suitable for publication in PLOS ONE. We carefully considered and effected all the insightful comments from the peer-reviewers, especially Reviewer #2. We hope that you will find our revised manuscript suitable for publication in your esteemed journal. For a detailed point-by-point response to reviewers, please see over.

Looking forward to hearing from you soon concerning an editorial decision.

Yours sincerely,

David Patrick Kateete Ph.D.

Makerere University College of Health Sciences

Kampala, Uganda

 

POINT-BY-POINT RESPONSE TO THE REVIEWERS’ COMMENTS 

SPECIFIC EDITORIAL COMMENT:

Both reviewers agreed that the revised manuscript show significant improvement, however, there are still some issues/concerns that need to be addressed. Please review and address all comments from reviewer #2.

RESPONSE:

We have fully effected all the comments from Reviewer #2. 

REVIEW COMMENTS TO THE AUTHOR:

COMMENT:

Reviewer #1: The authors have addressed all my suggestions. I found their responses quite satisfactory and the revised version has been much improved. I now recommend the paper for publication with no revisions.

RESPONSE:

We are delighted to hear so. Thank you for taking time to review the manuscript. 

COMMENT:

Reviewer #2: The addition of some extra information about the hospital strains was helpful – but it is still inadequate. The methods needs to include information of how these were obtained, who from, what diseases they caused etc. If they’re from adults then careful explanation needs to be made to explain why a study of colonising strains from children would be compared to strains from a hospital obtained from adults. The two strain collections don’t really add up – all staphs vs. S. aureus and S. epi only. (My personal opinion is that this comparison actually doesn’t add a great deal to your manuscript).

RESPONSE:

In the current revision, we have heeded your advice and removed the comparison between previously characterized hospital isolates and the current study. We thank you for the direction.

COMMENT:

Line 57 … S. aureus strains are overwhelmingly coagulase-positive, some are atypical in that they do not produce coagulase.

This requires a reference to be added – ideally several references.

RESPONSE:

References (See 5-11) have been added to this effect. Thank you.

COMMENT:

Line 91 … “with the assumption that hospital-associated isolates are more invasive/virulent and/or resistant to antibiotics compared to community-associated isolates.” Why did you have this assumption? – prior research has shown that most hospital acquired S. aureus infections are endogenous – caused by the carriage strain. This might best be stated as the null hypothesis.

RESPONSE:

In this revision, we heeded your advice and removed the comparison between previously characterized hospital isolates and the isolates in the current study as it was confusing. Thank you. 

COMMENT:

Table 4 still contains P values with a lot of decimal points, as does the text. Use 1 d.p. or use <0.05; <0.001. “S. haemolyticus was significantly more resistant to cefoxitin (p=0.0055), tetracycline (p=0.0041), SXT (p<0.0001), erythromycin (p<0.0001)…” should become: “S. haemolyticus was significantly more resistant to cefoxitin (p<0.05), tetracycline (p<0.05), SXT (p<0.001), erythromycin (p<0.001)”. These P values are low – but they don’t help the reader understand the message. It would be better to provide the proportions. These are NOT provided in table 1. So the text would read: “Resistance among S. aureus isolates to cefoxitin (x/y, z%), tetracycline (x/y, z%)…..was less common than S. haem isolates (cefoxitin a/b, c% resistant; tetracycline….). This type of information would be better in a table. Again – I don’t think the chi-sq testing is useful.

RESPONSE:

We have revised the text as advised. We have provided proportions, and removed the p-values from the results as they are inconsequential. This information is summarized in Table 3, which we have cited in the results section. We hope that the text is now easier to understand. We thank you.

---

## [Decision Letter · Decision Letter 2]

29 Jan 2020

Species and drug susceptibility profiles of staphylococci isolated from healthy children in Eastern Uganda

PONE-D-19-21368R2

Dear Dr. Kateete,

We are pleased to inform you that your manuscript has been judged scientifically suitable for publication and will be formally accepted for publication once it complies with all outstanding technical requirements.

With kind regards,

Baochuan Lin, Ph.D.

Academic Editor

PLOS ONE

Additional Editor Comments (optional):

Reviewers' comments:

Reviewer's Responses to Questions

**Comments to the Author**

1. If the authors have adequately addressed your comments raised in a previous round of review and you feel that this manuscript is now acceptable for publication, you may indicate that here to bypass the “Comments to the Author” section, enter your conflict of interest statement in the “Confidential to Editor” section, and submit your "Accept" recommendation.

Reviewer #2: All comments have been addressed

2. Is the manuscript technically sound, and do the data support the conclusions?

Reviewer #2: Yes

3. Has the statistical analysis been performed appropriately and rigorously? 

Reviewer #2: Yes

4. Have the authors made all data underlying the findings in their manuscript fully available?

Reviewer #2: Yes

5. Is the manuscript presented in an intelligible fashion and written in standard English?

Reviewer #2: Yes

6. Review Comments to the Author

Reviewer #2: Thank you - I found this revised paper to be more succinct and streamlined. The parts that were confusing have been removed.

7. PLOS authors have the option to publish the peer review history of their article (what does this mean?). If published, this will include your full peer review and any attached files.

Reviewer #2: Yes: S R Ritchie

---

## [Editor Report · Acceptance letter]

5 Feb 2020

PONE-D-19-21368R2 

Species and drug susceptibility profiles of staphylococci isolated from healthy children in Eastern Uganda 

Dear Dr. Kateete:

I am pleased to inform you that your manuscript has been deemed suitable for publication in PLOS ONE. Congratulations! Your manuscript is now with our production department. 

With kind regards,

on behalf of

Dr. Baochuan Lin 

Academic Editor

PLOS ONE